# A STUDY OF FACE OBFUSCATION IN IMAGENET

## ABSTRACT

Face obfuscation (blurring, mosaicing, etc.) has been shown to be effective for privacy protection; nevertheless, object recognition research typically assumes access to complete, unobfuscated images. In this paper, we explore the effects of face obfuscation on the popular ImageNet challenge visual recognition benchmark. Most categories in the ImageNet challenge are not people categories; however, many incidental people appear in the images, and their privacy is a concern. We first annotate faces in the dataset. Then we demonstrate that face blurring and overlaying—two typical obfuscation techniques—have minimal impact on the accuracy of recognition models. Concretely, we benchmark multiple deep neural networks on face-obfuscated images and observe that the overall recognition accuracy drops only slightly ($\leq 1.0\%$). Further, we experiment with transfer learning to 4 downstream tasks (object recognition, scene recognition, face attribute classification, and object detection) and show that features learned on face-obfuscated images are equally transferable. Our work demonstrates the feasibility of privacy-aware visual recognition, improves the highly-used ImageNet challenge benchmark, and suggests an important path for future visual datasets.

## 1 INTRODUCTION

Visual data is being generated at an unprecedented scale. People share billions of photos daily on social media (Meeker, 2014). There is one security camera for every 4 people in China and the United States (Lin & Purnell, 2019). Even your home can be watched by smart devices taking photos (Butler et al., 2015; Dai et al., 2015). Learning from the visual data has led to computer vision applications that promote the common good, e.g., better traffic management (Malhi et al., 2011) and law enforcement (Sajjad et al., 2020). However, it also raises privacy concerns, as images may capture sensitive information such as faces, addresses, and credit cards (Orekondy et al., 2018).

Extensive prior research has focused on preventing unauthorized access to sensitive information in private datasets (Fredrikson et al., 2015; Shokri et al., 2017). However, *are publicly available datasets free of privacy concerns?* Taking the popular ImageNet dataset (Deng et al., 2009) as an example, there are only 3 people categories[1] in the 1000 categories of the ImageNet Large Scale Visual Recognition Challenge (ILSVRC) (Russakovsky et al., 2015); nevertheless, the dataset exposes many people co-occurring with other objects in images (Prabhu & Birhane, 2021), e.g., people sitting on chairs, walking dogs, or drinking beer (Fig. 1). It is concerning since ILSVRC is freely available for academic use[2] and widely used by the research community.

In this paper, we attempt to mitigate ILSVRC's privacy issues. Specifically, we construct a privacy-enhanced version of ILSVRC and gauge its utility as a benchmark for image classification and as a dataset for transfer learning.

**Face annotation.** As an initial step, we focus on a prominent type of private information—faces. To examine and mitigate their privacy issues, we first annotate faces in ImageNet using face detectors and crowdsourcing. We use Amazon Rekognition to detect faces automatically, and then refine the results through crowdsourcing on Amazon Mechanical Turk to obtain accurate annotations.

We have annotated 1,431,093 images in ILSVRC, resulting in 562,626 faces from 243,198 images (17% of all images have at least one face). Many categories have more than 90% images with faces,

---

[1] `scuba diver`, `bridegroom`, and `baseball player`
[2] https://image-net.org/request

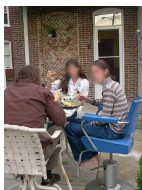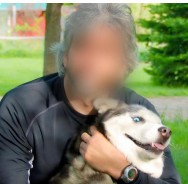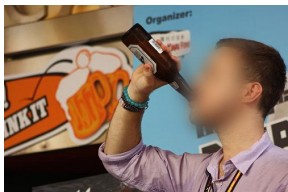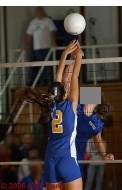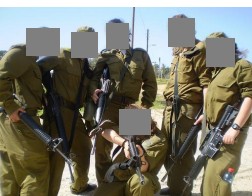

Figure 1: Most categories in ImageNet Challenge (Russakovsky et al., 2015) are not people categories. However, the images contain many people co-occurring with the object of interest, posing a potential privacy threat. These are example images (with faces blurred or overlaid) of `barber chair`, `husky`, `beer bottle`, `volleyball` and `military uniform`.

even though they are not people categories, e.g., `volleyball` and `military uniform`. Our annotations confirm that faces are ubiquitous in ILSVRC and pose a privacy issue. We release the face annotations to facilitate subsequent research in privacy-aware visual recognition on ILSVRC.

**Effects of face obfuscation on classification accuracy.** Obfuscating sensitive image areas is widely used for preserving privacy (McPherson et al., 2016). We focus on two simple obfuscation methods: blurring and overlaying (Fig. 1), whose privacy effects have been analyzed in prior work (Oh et al., 2016; Li et al., 2017; Hasan et al., 2018). Using our face annotations, we construct face-obfuscated versions of ILSVRC. *What are the effects of using them for image classification?* At first glance, it seems inconsequential—one should still recognize a car even when the people inside have their faces blurred. Indeed, we verify that validation accuracy drops only slightly (0.1%–0.7% for blurring, 0.3%–1.0% for overlaying) when using face-obfuscated images to train and evaluate. We analyze this drop in detail (identifying categories which are particularly affected), but this key result demonstrates that we can train privacy-aware visual classifiers on ILSVRC which remain highly competitive, with less than a 1% accuracy drop.

**Effects on feature transferability.** Besides a classification benchmark, ILSVRC also serves as pretraining data for transferring to domains where labeled images are scarce (Girshick, 2015; Liu et al., 2015a). So a further question is: *Does face obfuscation hurt the transferability of visual features learned from ILSVRC?* We investigate by pretraining models on the original/obfuscated images and finetuning on 4 downstream tasks: object recognition on CIFAR-10 (Krizhevsky et al., 2009), scene recognition on SUN (Xiao et al., 2010), object detection on PASCAL VOC (Everingham et al., 2010), and face attribute classification on CelebA (Liu et al., 2015b). They include both classification and spatial localization, as well as both face-centric and face-agnostic recognition. In all of the 4 tasks, models pretrained on face-obfuscated images perform closely with models pretrained on original images. We do not see a statistically significant difference between them, suggesting that visual features learned from face-obfuscated pretraining are equally transferable. Again, this encourages us to adopt face obfuscation as an additional protection on visual recognition datasets without worrying about detrimental effects on the dataset's utility.

**Contributions.** Our contributions are twofold. First, we obtain accurate face annotations in ILSVRC, facilitating subsequent research on privacy protection. We will release the code and the annotations. Second, to the best of our knowledge, we are the first to investigate the effects of privacy-aware face obfuscation on large-scale visual recognition. Through extensive experiments, we demonstrate that training on face-obfuscated images does not significantly compromise accuracy on both image classification and downstream tasks, while providing some privacy protection. Therefore, we advocate for face obfuscation to be included in ImageNet and to become a standard step in future dataset creation efforts.

## 2    RELATED WORK

**Privacy-preserving machine learning (PPML).** Machine learning frequently uses private datasets (Chen et al., 2019b). Research in PPML is concerned with an adversary trying to infer the private data. The privacy breach can happen to the trained model. For example, *model inversion attack* recovers sensitive attributes (e.g., gender, genotype) of an individual given the model's output (Fredrikson et al., 2014; 2015; Hamm, 2017; Li et al., 2019; Wu et al., 2019). *Membership inference attack* infers whether an individual was included in training (Shokri et al., 2017; Nasr et al., 2019; Hisamoto et al., 2020). *Training data extraction attack* extracts verbatim training data from the model (Carlini et al., 2019; 2020). For defending against these attacks, *differential privacy* is a general framework (Abadi et al., 2016; Chaudhuri & Monteleoni, 2008; McMahan et al., 2018; Jayaraman & Evans, 2019; Jagielski et al., 2020). It requires the model to behave similarly whether or not an individual is in the training data.

Privacy breaches can also happen in training/inference. To address hardware/software vulnerabilities, researchers have used *enclaves*—a hardware mechanism for protecting a memory region from unauthorized access—to execute machine learning workloads (Ohrimenko et al., 2016; Tramer & Boneh, 2018). Machine learning service providers can run their models on users' private data encrypted using *homomorphic encryption* (Gilad-Bachrach et al., 2016; Brutzkus et al., 2019; Juvekar et al., 2018; Bian et al., 2020; Yonetani et al., 2017). It is also possible for multiple data owners to train a model collectively without sharing their private data using federated learning (McMahan et al., 2017; Bonawitz et al., 2017; Li et al., 2020) or secure multi-party computation (Shokri & Shmatikov, 2015; Melis et al., 2019; Hamm et al., 2016; Pathak et al., 2010; Hamm et al., 2016).

There is a fundamental difference between our work and PPML. PPML focuses on private datasets, whereas we focus on public datasets with private information. ImageNet, like other academic datasets, is publicly available to researchers. There is no point preventing an adversary from inferring the data. However, public datasets can also expose private information about individuals, who may not even be aware of their presence in the data. It is their privacy we are protecting.

**Privacy in visual data.** To mitigate privacy issues with public visual datasets, researchers have attempted to obfuscate private information before publishing the data. Frome et al. (2009) and Uittenbogaard et al. (2019) use blurring and inpainting to obfuscate faces and license plates in Google Street View. nuScenes (Caesar et al., 2020) is an autonomous driving dataset where faces and license plates are detected and then blurred. Similar method is also used for the action dataset AViD (Piergiovanni & Ryoo, 2020).

We follow this line of work to obfuscate faces in ImageNet but differ in two critical ways. First, to the best of our knowledge, we are the first to thoroughly analyze the effects of face obfuscation on visual recognition. Second, prior works use only automatic methods such as face detectors, whereas we additionally employ crowdsourcing. Human annotations are more accurate and thus more useful for following research on privacy preservation in ImageNet. Most importantly though, automated face recognition methods are known to contain racial and gender biases (Buolamwini & Gebru, 2018); thus using these methods alone is likely to result in more privacy protection to members of majority groups. Including a manual verification step helps partially mitigate these issues.

Finally, we note that face obfuscation alone is not sufficient for privacy protection. Orekondy et al. (2018) constructed Visual Redactions, annotating images with 42 privacy attributes, including faces, names, and addresses. Ideally, we should obfuscate all such information; however, this may not be immediately feasible. Obfuscating faces (omnipresent in visual datasets) is an important first step.

**Privacy guarantees of face obfuscation.** Unfortunately, face obfuscation does not provide any formal guarantee of privacy. Both humans and machines may be able to infer an individual's identity from face-obfuscated images, presumably relying on cues outside faces such as height and clothing (Chang et al., 2006; Oh et al., 2016). Researchers have tried to protect sensitive image regions against attacks, e.g., by perturbing the image adversarially to reduce the performance of a recognizer (Oh et al., 2017; Ren et al., 2018; Sun et al., 2018; Wu et al., 2018; Xiao et al., 2020). However, these methods are tuned for a particular model and provide no privacy guarantee either.

Further, guarantees in privacy may reduce dataset utility as shown for example by Cheng et al. (2021). Therefore, we choose two simple local methods—blurring and overlaying—instead of more sophisticated alternatives. Overlaying removes all information in a face bounding box, whereas blurring removes only partial information. Their effectiveness for privacy protection can be ascertained only empirically, which has been the focus of prior work (Oh et al., 2016; Li et al., 2017; Hasan et al., 2018) but is beyond the scope of this paper.

**Visual recognition from degraded data.** Researchers have studied visual recognition in the presence of various image degradation, including blurring (Vasiljevic et al., 2016), lens distortions (Pei et al., 2018), and low resolution (Ryoo et al., 2016). These undesirable artifacts are due to imperfect sensors rather than privacy concerns. In contrast, we intentionally obfuscate faces for privacy's sake.

**Ethical issues with datasets.** Datasets are important in machine learning and computer vision. But recently they have been called out for scrutiny (Paullada et al., 2020), especially regarding the presence of people. A prominent issue is imbalanced representation, e.g., underrepresentation of certain demographic groups in data for face recognition (Buolamwini & Gebru, 2018), activity recognition (Zhao et al., 2017), and image captioning (Hendricks et al., 2018).

For ImageNet, researchers have examined and attempted to mitigate issues such as geographic diversity, the category vocabulary, and imbalanced representation (Shankar et al., 2017; Stock & Cisse, 2018; Dulhanty & Wong, 2019; Yang et al., 2020). We focus on an orthogonal issue: the privacy of people in the images. Prabhu & Birhane (2021) also discussed ImageNet's privacy issues and suggested face obfuscation as one potential solution. Our face annotations enable face obfuscation to be implemented, and our experiments support its effectiveness. Concurrent work (Asano et al., 2021) addresses the privacy issue by collecting a dataset of unlabeled images without people.

**Potential negative impacts.** The main concern we see is giving the impression of privacy *guarantees* when in fact face obfuscation is an imperfect technique for privacy protection. We hope that the above detailed discussion and this clarification will help mitigate this issue. Another important concern is disparate impact on people of different demographics as a result of using automated face detection methods; as mentioned above, we hope that incorporating a manual annotation step will help partially alleviate this issue so that similar privacy preservation is afforded to all.

## 3 ANNOTATING FACES IN ILSVRC

We annotate faces in ILSVRC (Russakovsky et al., 2015). The annotations localize an important type of sensitive information in ImageNet, making it possible to obfuscate the sensitive areas for privacy protection.

It is challenging to annotate faces accurately, at ImageNet's scale while under a reasonable budget. Automatic face detectors are fast and cheap but not accurate enough, whereas crowdsourcing is accurate but more expensive. Inspired by prior work (Kuznetsova et al., 2018; Yu et al., 2015), we devise a two-stage semi-automatic pipeline that brings the best of both worlds. First, we run the face detector by Amazon Rekognition on all images in ILSVRC. The results contain both false positives and false negatives, so we refine them through crowdsourcing on Amazon Mechanical Turk. Workers are given images with detected bounding boxes, and they adjust existing boxes or create new ones to cover all faces. Please see Appendix A for detail.

**Annotation quality.** To analyze the quality of the face annotations, we select 20 categories on which the face detector is likely to perform poorly. Then we manually check validation images from these categories; the results characterize an upper bound of the overall annotation accuracy.

Concretely, first, we randomly sample 10 categories under the `mammal` subtree in the ImageNet hierarchy (the left 10 categories in Table 1). Images in these categories contain many false positives (animal faces detected as humans). Second, we take the 10 categories with the greatest number of detected faces (the right 10 categories in Table 1). Images in those categories contain many people and thus are likely to have more false negatives. Each of the selected categories has 50 validation images, and two graduate students manually inspected all face annotations on them, including the face detection results and the final crowdsourcing results.

Table 1: The number of false positives (*FPs*) and false negatives (*FNs*) on validation images from 20 categories challenging for the face detector. Each category has 50 images. The *A* columns are after automatic face detection, whereas the *H* columns are human results after crowdsourcing.

| Category | #FPs | | #FNs | | Category | #FPs | | #FNs | |
|---|---|---|---|---|---|---|---|---|---|
| | A | H | A | H | | A | H | A | H |
| irish setter | 12 | 3 | 0 | 0 | maypole | 0 | 0 | 7 | 5 |
| gorilla | 32 | 7 | 0 | 0 | basketball | 0 | 0 | 7 | 2 |
| cheetah | 3 | 1 | 0 | 0 | volleyball | 0 | 0 | 10 | 5 |
| basset | 10 | 0 | 0 | 0 | balance beam | 0 | 0 | 9 | 5 |
| lynx | 9 | 1 | 0 | 0 | unicycle | 0 | 1 | 6 | 1 |
| rottweiler | 11 | 4 | 0 | 0 | stage | 0 | 0 | 0 | 0 |
| sorrel | 2 | 1 | 0 | 0 | torch | 2 | 1 | 1 | 1 |
| impala | 1 | 0 | 0 | 0 | baseball player | 0 | 0 | 0 | 0 |
| bernese mt. dog | 20 | 3 | 0 | 0 | military uniform | 3 | 2 | 2 | 0 |
| silky terrier | 4 | 0 | 0 | 0 | steel drum | 1 | 1 | 1 | 0 |
| Average | 10.4 | 2.0 | 0.0 | 0.0 | Average | 0.6 | 0.5 | 4.3 | 1.9 |

Table 2: Some categories grouped into supercategories in WordNet (Miller, 1998). For each supercategory, we show the fraction of images with faces. These supercategories have fractions significantly deviating from the average of the entire ILSVRC (17%).

| Supercategory | #Categories | #Images | With faces (%) |
|---|---|---|---|
| clothing | 49 | 62,471 | 58.90 |
| wheeled vehicle | 44 | 57,055 | 35.30 |
| musical instrument | 26 | 33,779 | 47.64 |
| bird | 59 | 76,536 | 1.69 |
| insect | 27 | 35,097 | 1.81 |

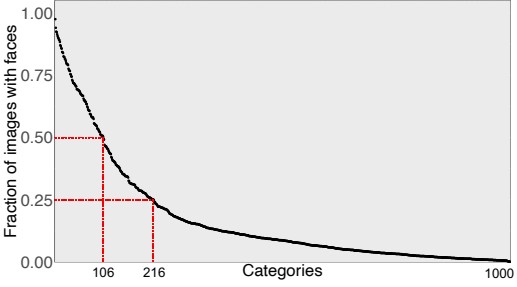 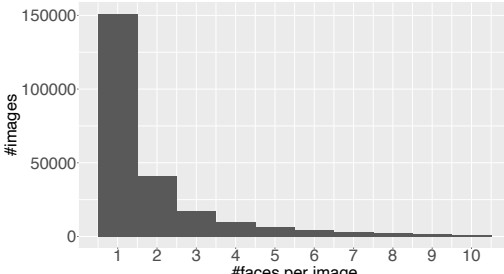

Figure 2: *Left*: The fraction of images with faces for the 1000 ILSVRC categories. 106 categories have more than half images with faces. 216 categories have more than 25%. *Right*: A histogram of the number of faces per image, excluding the 1,187,895 images with no face.

The errors are shown in Table 1. As expected, the left 10 categories (mammals) have some false positives but no false negatives. In contrast, the right 10 categories have very few false positives but some false negatives. Crowdsourcing significantly reduces both error types. This demonstrate that we can obtain high-quality face annotations using the two-stage pipeline, but face detection alone is less accurate. Among the 20 categories, we have on average 1.25 false positives and 0.95 false negatives per 50 images. However, our overall accuracy on the entire ILSVRC is much higher as these categories are selected deliberately to be error-prone.

**Distribution of faces in ILSVRC.** Using our two-stage pipeline, we annotated all 1,431,093 images in ILSVRC. Among them, 243,198 images (17%) contain at least one face. And the total number of faces adds up to 562,626.

Fig. 2 *Left* shows the fraction of images with faces for different categories, ranging from 97.5% (bridegroom) to 0.1% (rock beauty, a type of saltwater fish). 106 categories have more than half images with faces. 216 categories have more than 25%. Among the 243K images with faces, Fig. 2 *Right* shows the number of faces per image. 90.1% images contain less than 5. But some of them contain as many as 100 (a cap due to Amazon Rekognition). Most of those images capture sports scenes with a crowd of spectators, e.g., images from baseball player or volleyball.

Since ILSVRC categories are in the WordNet (Miller, 1998) hierarchy, we can group them into supercategories in WordNet. Table 2 lists a few common ones that collectively cover 215 categories. For each supercategory, we calculate the fraction of images with faces. Results suggests that supercategories such as clothing and musical instrument frequently co-occur with people, whereas bird and insect seldom do.

## 4 EFFECTS OF FACE OBFUSCATION ON CLASSIFICATION ACCURACY

Having annotated faces in ILSVRC, we now investigate how face obfuscation—a widely used technique for privacy preservation (Fan, 2019; Frome et al., 2009)—impacts image classification.

**Face obfuscation method.** We experiment with two simple obfuscation methods—blurring and overlaying. For overlaying, we cover faces with the average color in the ILSVRC training data: a gray shade with RGB value $(0.485, 0.456, 0.406)$. For blurring, we use a variant of Gaussian blurring. It achieves better visual quality by removing the sharp boundaries between blurred and unblurred regions (Fig. 1). Let $I$ be an image and $M$ be the mask of face bounding boxes. Applying Gaussian blurring to them gives us $I_{blurred}$ and $M_{blurred}$. Then we use $M_{blurred}$ as the mask to composite $I$ and $I_{blurred}$: $I_{new} = M_{blurred} \cdot I_{blurred} + (1 - M_{blurred}) \cdot I$. Due to the use of $M_{blurred}$ instead of $M$, we avoid sharp boundaries in $I_{new}$. Please see Appendix B for detail.

Table 3: Validation accuracies on ILSVRC using original images, face-blurred images, and face-overlaid images. The accuracy drops slightly but consistently when blurred (the $\Delta_b$ columns) or overlaid (the $\Delta_o$ columns), though overlaying leads to larger drop than blurring. Each experiment is repeated 3 times; we report the mean accuracy and its standard error (SEM).

| Model | Top-1 accuracy (%) | | | | | Top-5 accuracy (%) | | | | |
|---|---|---|---|---|---|---|---|---|---|---|
| | Original | Blurred | $\Delta_b$ | overlaid | $\Delta_o$ | Original | Blurred | $\Delta_b$ | overlaid | $\Delta_o$ |
| AlexNet | $56.0 \pm 0.3$ | $55.8 \pm 0.1$ | 0.2 | $55.5 \pm 0.2$ | 0.6 | $78.8 \pm 0.1$ | $78.6 \pm 0.1$ | 0.3 | $78.2 \pm 0.2$ | 0.7 |
| SqueezeNet | $56.0 \pm 0.2$ | $55.3 \pm 0.0$ | 0.7 | $55.0 \pm 0.2$ | 1.0 | $78.6 \pm 0.2$ | $78.1 \pm 0.0$ | 0.5 | $77.6 \pm 0.1$ | 1.0 |
| ShuffleNet | $64.7 \pm 0.2$ | $64.0 \pm 0.1$ | 0.6 | $63.7 \pm 0.0$ | 1.0 | $85.9 \pm 0.0$ | $85.5 \pm 0.1$ | 0.5 | $85.2 \pm 0.2$ | 0.8 |
| VGG11 | $68.9 \pm 0.0$ | $68.2 \pm 0.1$ | 0.7 | $67.8 \pm 0.2$ | 1.1 | $88.7 \pm 0.0$ | $88.3 \pm 0.1$ | 0.4 | $87.9 \pm 0.0$ | 0.8 |
| VGG13 | $69.9 \pm 0.1$ | $69.3 \pm 0.1$ | 0.7 | $68.8 \pm 0.0$ | 1.2 | $89.3 \pm 0.1$ | $88.9 \pm 0.1$ | 0.4 | $88.5 \pm 0.1$ | 0.8 |
| VGG16 | $71.7 \pm 0.1$ | $70.8 \pm 0.1$ | 0.8 | $70.6 \pm 0.1$ | 1.1 | $90.5 \pm 0.1$ | $89.9 \pm 0.1$ | 0.6 | $89.6 \pm 0.0$ | 0.9 |
| VGG19 | $72.4 \pm 0.0$ | $71.5 \pm 0.0$ | 0.8 | $71.2 \pm 0.2$ | 1.2 | $90.9 \pm 0.1$ | $90.3 \pm 0.0$ | 0.6 | $90.1 \pm 0.1$ | 0.8 |
| MobileNet | $65.4 \pm 0.2$ | $64.4 \pm 0.2$ | 1.0 | $64.3 \pm 0.2$ | 1.0 | $86.7 \pm 0.1$ | $86.0 \pm 0.1$ | 0.7 | $85.7 \pm 0.1$ | 0.9 |
| DenseNet121 | $75.0 \pm 0.1$ | $74.2 \pm 0.1$ | 0.8 | $74.1 \pm 0.1$ | 1.0 | $92.4 \pm 0.0$ | $92.0 \pm 0.1$ | 0.4 | $91.7 \pm 0.0$ | 0.7 |
| DenseNet201 | $77.0 \pm 0.0$ | $76.6 \pm 0.0$ | 0.4 | $76.1 \pm 0.1$ | 0.9 | $93.5 \pm 0.0$ | $93.2 \pm 0.1$ | 0.2 | $92.9 \pm 0.1$ | 0.6 |
| ResNet18 | $69.8 \pm 0.2$ | $69.0 \pm 0.2$ | 0.7 | $68.9 \pm 0.1$ | 0.8 | $89.2 \pm 0.0$ | $88.7 \pm 0.0$ | 0.5 | $88.7 \pm 0.1$ | 0.6 |
| ResNet34 | $73.1 \pm 0.1$ | $72.3 \pm 0.4$ | 0.8 | $72.4 \pm 0.1$ | 0.7 | $91.3 \pm 0.0$ | $90.8 \pm 0.1$ | 0.5 | $90.7 \pm 0.0$ | 0.6 |
| ResNet50 | $75.5 \pm 0.2$ | $75.0 \pm 0.1$ | 0.4 | $74.9 \pm 0.0$ | 0.6 | $92.5 \pm 0.0$ | $92.4 \pm 0.1$ | 0.1 | $92.2 \pm 0.0$ | 0.3 |
| ResNet101 | $77.3 \pm 0.1$ | $76.7 \pm 0.1$ | 0.5 | $76.7 \pm 0.1$ | 0.6 | $93.6 \pm 0.1$ | $93.3 \pm 0.1$ | 0.3 | $93.1 \pm 0.1$ | 0.5 |
| ResNet152 | $77.9 \pm 0.1$ | $77.3 \pm 0.1$ | 0.6 | $77.0 \pm 0.3$ | 0.9 | $93.9 \pm 0.0$ | $93.7 \pm 0.0$ | 0.4 | $93.3 \pm 0.3$ | 0.6 |
| Average | **70.0** | 69.4 | 0.7 | 69.1 | 0.9 | **89.1** | 88.6 | 0.4 | 88.4 | 0.7 |

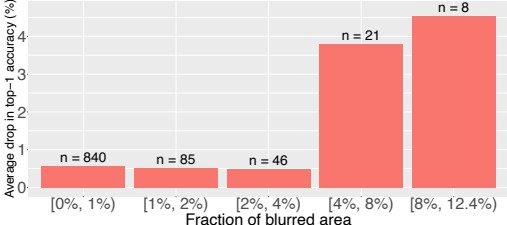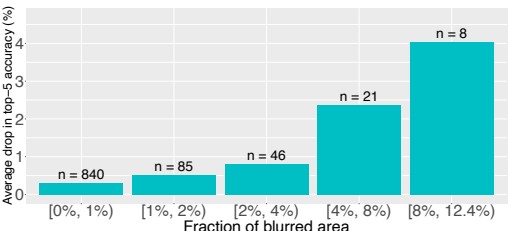

Figure 3: The average drop in category-wise accuracies vs. the fraction of blurred area in images. *Left*: Top-1 accuracies. *Right*: Top-5 accuracies. The accuracies are averaged across all different model architectures and random seeds.

**Experiment setup and training details.** To study the effects of face obfuscation on classification, we benchmark various deep neural networks including AlexNet (Krizhevsky et al., 2017), VGG (Simonyan & Zisserman, 2015), SqueezeNet (Iandola et al., 2016), ShuffleNet (Zhang et al., 2018), MobileNet (Howard et al., 2017), ResNet (He et al., 2016), and DenseNet (Huang et al., 2017). Each model is studied in three settings: (1) original images for both training and evaluation; (2) face-blurred images for both; (3) face-overlaid images for both.

Different models share a uniform implementation of the training/evaluation pipeline. During training, we randomly sample a $224 \times 224$ image crop and apply random horizontal flipping. During evaluation, we always take the central crop and do not flip. All models are trained with a batch size of 256, a momentum of 0.9, and a weight decay of $10^{-4}$. We train with SGD for 90 epochs, dropping the learning rate by a factor of 10 every 30 epochs. The initial learning rate is 0.01 for AlexNet, SqueezeNet, and VGG; 0.1 for other models. Each experiment takes 1–7 days on machines with 2 CPUs, 16GB memory, and 1–6 Nvidia GTX GPUs.

**Overall accuracy.** Table 3 shows the validation accuracies. Each training instance is replicated 3 times with different random seeds, and we report the mean accuracy and its standard error (SEM). The $\Delta$ columns are the accuracy drop when using face-obfuscated images (original minus blurred). For both blurring and overlaying, we see a small but consistent drop in top-1 and top-5 accuracies. For example, with blurring, top-5 accuracies drop 0.1%–0.7% with an average of only $0.4\%$. Overlaying leads to slightly larger drops averaged at $0.7\%$ since it removes more information.

It is expected to incur a small but consistent drop. On the one hand, face obfuscation removes information that might be useful for classifying the image. On the other hand, it should leave intact most ILSVRC categories since they are non-human. Though not surprising, our results are encouraging. They assure us that we can train privacy-aware visual classifiers on ImageNet with less than 1% accuracy drop.

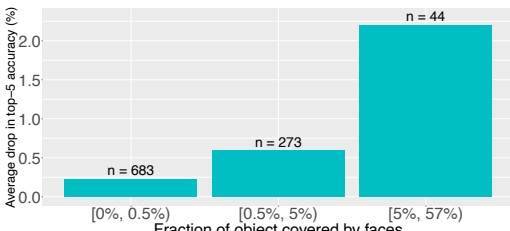

Figure 4: The average drop in category-wise accuracies caused by blurring vs. the fraction of object area covered by faces. *Left*: Top-1 accuracies. *Right*: Top-5 accuracies. The accuracies are averaged across all different model architectures and random seeds.

**Category-wise accuracies and the fraction of blur.** To gain insights into the effects on individual categories, we break down the accuracy into the 1000 ILSVRC categories. We hypothesize that if a category has a large fraction of obfuscated area, it will likely incur a large accuracy drop.

To support the hypothesis, we focus on blurring and first average the accuracies for each category across different models. Then, we calculate the correlation between the accuracy drop and the fraction of blurred area: $r = 0.28$ for top-1 accuracy and $r = 0.44$ for top-5 accuracy. The correlation is not strong but is statistically significant, with p-values of $6.31 \times 10^{-20}$ and $2.69 \times 10^{-49}$ respectively.

The positive correlation is also evident in Fig. 3. On the x-axis, we divide the blurred fraction into 5 groups from small to large. On the y-axis, we show the average accuracy drop for categories in each group. Using top-5 accuracy (Fig. 3 *Right*), the drop increases monotonically from 0.30% to 4.04% when moving from a small blurred fraction (0%–1%) to a larger fraction ($\geq 8\%$).

The pattern becomes less clear in top-1 accuracy (Fig. 3 *Left*). The drop stays around $0.5\%$ and begins to increase only when the fraction goes beyond 4%. However, top-1 accuracy is a worse metric than top-5 accuracy (ILSVRC's official metric), because top-1 accuracy is ill-defined for images with multiple objects. In contrast, top-5 accuracy allows the model to predict 5 categories for each image and succeed as long as one of them matches the ground truth. In addition, top-1 accuracy suffers from confusion between near-identical categorie (like `eskimo dog` and `siberian husky`), an artifact we discuss further below.

In summary, our analysis of category-wise accuracies aligns with a simple intuition—if too much area is obfuscated, models will have difficulty classifying the image.

**Most impacted categories.** Besides the size of the obfuscated area, another factor is whether it overlaps with the object of interest. Most categories in ILSVRC are non-human and should have very little overlap with faces. However, there are exceptions. `Mask`, for example, is indeed non-human. But masks are worn on the face; therefore, obfuscating faces will make masks harder to recognize. Similar categories include `sunglasses`, `harmonica`, etc. Due to their close spatial proximity to faces, the accuracy is likely to drop significantly in the presence of face obfuscation.

To quantify this intuition, we calculate the overlap between objects and faces. Object bounding boxes are available from the localization task of ILSVRC. Given an object bounding box, we calculate the fraction of area covered by face bounding boxes. The fractions are then averaged across different images in a category.

Results in Fig. 4 show that blurring leads to larger accuracy drop for categories with larger fractions covered by faces. Some notable examples include `mask` (24.84% covered by faces, 8.71% drop in top-5 accuracy), `harmonica` (29.09% covered by faces, 8.93% drop in top-5 accuracy), and `snorkel` (30.51% covered, 6.00% drop). The correlation between the fraction and the drop is $r = 0.32$ for top-1 accuracy and $r = 0.46$ for top-5 accuracy.

Fig. 5a showcases images from `harmonica` and `mask` and their blurred versions. We use Grad-CAM (Selvaraju et al., 2017) to visualize where the model is looking at when classifying the image. For original images, the model can effectively localize and classify the object of interest. For blurred images, however, the model fails to classify the object; neither does it attend to the correct region.

In summary, the categories most impacted by face obfuscation are those overlapping with faces, such as `mask` and `harmonica`. These categories have much lower accuracies when using obfuscated images, as obfuscation removes visual cues necessary for recognizing them.

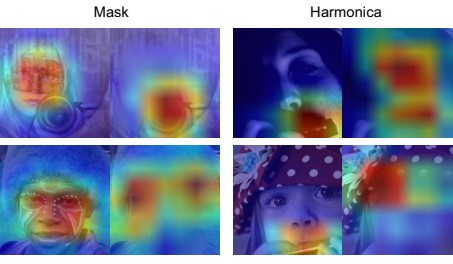
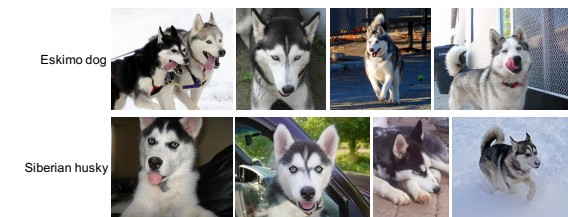

(a) Images from `mask` and `harmonica` with Grad-CAM (Selvaraju et al., 2017) visualizations of where a ResNet152 (He et al., 2016) model looks at. Original images on the left; face-blurred images on the right.

(b) Images from `eskimo dog` and `siberian husky` are very similar. However, `eskimo dog` has a large accuracy drop when using face-blurred images, whereas `siberian husky` has a large accuracy increase.

**Disparate changes for visually similar categories.** Our last observation focuses on categories whose top-1 accuracies change drastically. Intriguingly, they come in pairs, consisting of one category with decreasing accuracy and another visually similar category with increasing accuracy. For example, `eskimo dog` and `siberian husky` are visually similar (Fig. 5b). When using face-blurred images, `eskimo dog`'s top-1 accuracy drops by 12.8%, whereas `siberian husky`'s increases by 16.9%. It is strange since most images in these two categories do not even contain human faces. More examples are in Table 4.

`Eskimo dog` and `siberian husky` images are so similar that the model faces a seemingly arbitrary choice. We examine the predictions and find that models trained on original images prefer `eskimo dog`, whereas models trained on blurred images prefer `siberian husky`. It is the different preferences over these two competing categories that drive the top-1 accuracies to change in different directions. To further investigate, we include two metrics that are less sensitive to competing categories: top-5 accuracy and average precision. In Table 4, the pairwise pattern evaporates when these metrics. A pair of categories no longer have drastic changes, and the changes do not necessarily go in different directions. The results show that models trained on blurred images are still good at recognizing `eskimo dog`, though `siberian husky` has an even higher score.

## 5 EFFECTS ON FEATURE TRANSFERABILITY

Visual features learned on ImageNet are effective for a wide range of tasks (Girshick, 2015; Liu et al., 2015a). We now investigate the effects of face obfuscation on feature transferability to downstream tasks. Specifically, we compare models without pretraining and models pretrained on original/blurred/overlaid images by finetuning on 4 tasks: object recognition, scene recognition, object detection, and face attribute classification. They include both classification and spatial localization, as well as both face-centric and face-agnostic recognition. Details are in Appendix E.

**Object and scene recognition on CIFAR-10 and SUN.** CIFAR-10 (Krizhevsky et al., 2009) contains images from 10 object categories such as `horse` and `truck`. SUN (Xiao et al., 2010) contains images from 397 scenes such as `bedroom` and `restaurant`. Like ImageNet, they are not people-centered but may contain people.

We finetune models to classify images in these two datasets and show the results in Table 5. For both datasets, pretraining helps significantly; models pretrained on blurred or overlaid images perform closely with those pretrained on original images. The results show that visual features learned on face-obfsucated images have no problem transferring to face-agnostic downstream tasks.

**Object detection on PASCAL VOC.** Next, we finetune models for object detection on PASCAL VOC (Everingham et al., 2010). We choose it instead of COCO (Lin et al., 2014) because it is small enough to benefit from pretraining. We finetune a FasterRCNN (Ren et al., 2015) object detector with a ResNet50 backbone pretrained on original/blurred/overlaid images. The results do not show a significant difference between them (79.40 ± 0.31, 79.29 ± 0.22, and 79.39 ± 0.02 in mAP).

PASCAL VOC includes `person` as one of its 20 object categories. And one could hypothesize that the model detects people relying on face cues. However, we do not observe a performance drop in face-obfuscated pretraining, even considering the AP of the `person` category (84.40 ± 0.14 original, 84.80 ± 0.50 blurred, and 84.47 ± 0.05 overlaid).

Table 4: Visually similar categories whose top-1 accuracy varies significantly—but in opposite directions. However, the pattern evaporates when using top-5 accuracy or average precision.

| Category | Top-1 accuracy (%) | | | Top-5 accuracy (%) | | | Average precision (%) | | |
|---|---|---|---|---|---|---|---|---|---|
| | Original | Blurred | Δ | Original | Blurred | Δ | Original | Blurred | Δ |
| eskimo dog | **50.8** ± 1.1 | 38.0 ± 0.4 | 12.8 | 95.5 ± 0.4 | 95.1 ± 0.2 | 0.4 | 19.4 ± 0.8 | 19.9 ± 0.5 | − 0.5 |
| siberian husky | 46.3 ± 1.8 | **63.2** ± 0.8 | − 16.9 | 97.0 ± 0.4 | 97.2 ± 0.3 | -0.2 | 29.2 ± 0.3 | 29.6 ± 0.5 | − 0.4 |
| projectile | **35.6** ± 0.9 | 21.7 ± 1.0 | 13.9 | 86.2 ± 0.4 | 85.5 ± 0.4 | 0.7 | 23.1 ± 0.4 | 22.5 ± 0.5 | 0.6 |
| missile | 31.6 ± 0.7 | **45.8** ± 0.8 | − 14.2 | 81.5 ± 0.7 | 81.8 ± 0.4 | − 0.3 | 20.4 ± 0.3 | 21.1 ± 0.6 | − 0.7 |
| tub | **35.5** ± 1.5 | 27.9 ± 0.6 | 7.6 | **79.4** ± 0.6 | 75.6 ± 0.5 | 3.8 | **19.9** ± 0.4 | 18.8 ± 0.2 | 1.1 |
| bathtub | 35.4 ± 1.0 | **42.5** ± 0.4 | − 7.1 | 78.9 ± 0.3 | **80.8** ± 1.2 | − 1.9 | **27.4** ± 0.8 | 25.1 ± 0.6 | 2.3 |
| american chameleon | **63.0** ± 0.4 | 54.7 ± 1.2 | 8.3 | 97.0 ± 0.5 | 96.6 ± 0.5 | 0.4 | 40.0 ± 0.2 | 39.3 ± 0.5 | 0.7 |
| green lizard | 42.0 ± 0.6 | **45.6** ± 1.2 | − 3.6 | 91.3 ± 0.3 | 89.7 ± 0.2 | 1.6 | 22.6 ± 0.8 | 22.4 ± 0.1 | 0.2 |

Table 5: Top-1 accuracy on CIFAR-10 (Krizhevsky et al., 2009) and SUN (Xiao et al., 2010) of models without pretraining, pretrained on original images, and pretrained on blurred images.

| Model | CIFAR-10 | | | | SUN | | | |
|---|---|---|---|---|---|---|---|---|
| | No pretrain | Original | Blurred | Overlaid | No pretrain | Original | Blurred | Overlaid |
| AlexNet | 83.3 ± 0.2 | 90.6 ± 0.0 | 90.9 ± 0.0 | **91.1** ± 0.0 | 26.2 ± 0.6 | 46.3 ± 0.1 | **46.5** ± 0.1 | 46.2 ± 0.0 |
| ShuffleNet | 92.3 ± 0.3 | **95.7** ± 0.0 | 95.4 ± 0.1 | 95.2 ± 0.1 | 33.8 ± 0.7 | 51.2 ± 0.1 | 50.4 ± 0.3 | 49.3 ± 0.3 |
| ResNet18 | 92.8 ± 0.1 | 96.1 ± 0.1 | 96.1 ± 0.1 | 96.1 ± 0.1 | 36.9 ± 4.8 | 55.0 ± 0.2 | 55.0 ± 0.1 | 55.1 ± 0.1 |
| ResNet34 | 90.6 ± 0.9 | 96.9 ± 0.1 | 97.0 ± 0.0 | 97.1 ± 0.2 | 40.3 ± 0.4 | 57.8 ± 0.0 | 57.9 ± 0.1 | 57.8 ± 0.1 |

**Face attribute classification on CelebA.** But what if the downstream task is entirely about understanding faces? Will face-obfuscated pretraining fail? We explore this question by classifying face attributes on CelebA (Liu et al., 2015b). Given a headshot, the model predicts multiple face attributes such as `smiling` and `eyeglasses`.

CelebA is too large to benefit from pretraining, so we finetune on a subset of 5K images. Table 6 shows the results in mAP. There is a discrepancy between different models, so we add a few more models. But overall, blurred/overlaid pretraining performs competitively. This is remarkable given that the task relies heavily on faces. A possible reason is that the model only learns low-level face-agnostic features during pretraining and learns face features in finetuning.

In all of the 4 tasks, pretraining on face-obfuscated images does not hurt the transferability of the learned feature. It suggests that one could use face-obfuscated ILSVRC for pretraining without degrading the downstream task, even when the downstream task requires an understanding of faces.

Table 6: mAP of face attribute classification on CelebA (Liu et al., 2015b), using subset of 5K training images.

| Model | No pretrain | Original | Blurred | Overlaid |
|---|---|---|---|---|
| AlexNet | 41.8 ± 0.5 | **55.5** ± 0.7 | 50.7 ± 0.8 | 52.5 ± 0.4 |
| ShuffleNet | 36.5 ± 0.7 | **55.6** ± 1.2 | 52.5 ± 1.0 | 53.5 ± 1.4 |
| ResNet18 | 45.1 ± 1.0 | 51.7 ± 1.9 | 51.8 ± 1.0 | 52.0 ± 0.6 |
| ResNet34 | 49.4 ± 2.4 | 55.6 ± 2.4 | 56.5 ± 1.9 | 56.4 ± 2.3 |
| ResNet50 | 48.7 ± 1.3 | 42.8 ± 0.9 | 50.9 ± 2.7 | 50.4 ± 0.5 |
| VGG11 | 48.7 ± 0.3 | 56.0 ± 0.7 | 57.4 ± 0.6 | 58.1 ± 0.9 |
| VGG13 | 47.2 ± 0.8 | 58.4 ± 0.6 | 59.0 ± 0.5 | 58.2 ± 0.4 |
| MobileNet | 43.8 ± 0.2 | 49.4 ± 0.8 | 49.9 ± 1.3 | 49.6 ± 1.3 |

## 6 CONCLUSION

We explored how face obfuscation affects recognition accuracy on ILSVRC. We annotated faces in the dataset and benchmarked deep neural networks on images with faces blurred or overlaid. Experimental results demonstrate face obfuscation enhances privacy with minimal impact on accuracy.

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

## A  SEMI-AUTOMATIC FACE ANNOTATION

We describe our face annotation method in detail. It consists of two stages: face detection followed by crowdsourcing.

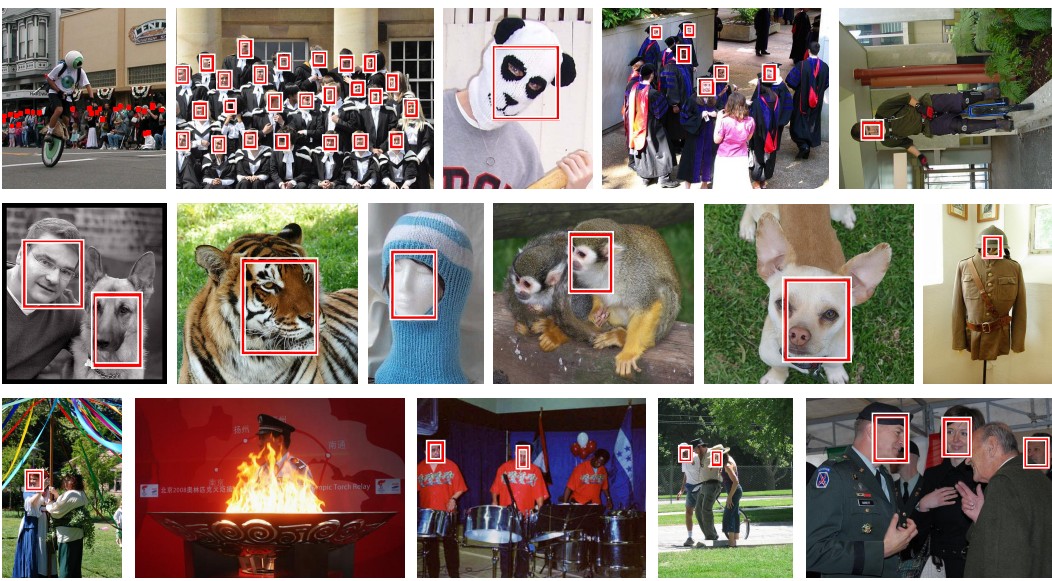

Figure 6: Face detection results on ILSVRC by Amazon Rekognition. The first row shows correct examples. The second row shows false positives, most of which are animal faces. The third row shows false negatives.

**Stage 1: Automatic face detection.**  First, we run the face detection API provided by Amazon Rekognition[3] on all images in ILSVRC, which can be done within one day and $1500. We also explored services from other vendors but found Rekognition to work the best, especially for small faces and multiple faces in one image (Fig. 6 *Top*).

However, face detectors are not perfect. There are false positives and false negatives. Most false positives, as Fig. 6 *Middle* shows, are animal faces incorrectly detected as humans. Meanwhile, false negatives are rare; some of them occur under poor lighting or heavy occlusion. For privacy preservation, a small number of false positives are acceptable, but false negatives are undesirable. In that respect, Rekognition hits a suitable trade-off for our purpose.

**Stage 2: Refining faces through crowdsourcing.**  After running the face detector, we refine the results through crowdsourcing on Amazon Mechanical Turk (MTurk). In each task, the worker is given an image with bounding boxes detected by the face detector (Fig. 7 *Left*). They adjust existing bounding boxes or create new ones to cover all faces and not-safe-for-work (NSFW) areas. NSFW areas may not necessarily contain private information, but just like faces, they are good candidates for image obfuscation (Prabhu & Birhane, 2021).

For faces, we specifically require the worker to cover the mouth, nose, eyes, forehead, and cheeks. For NSFW areas, we define them to include nudity, sexuality, profanity, etc. However, we do not dictate what constitutes, e.g., nudity, which is deemed to be subjective and culture-dependent. Instead, we encourage workers to follow their best judgment.

The worker has to go over 50 images in each HIT (Human Intelligence Task) to get rewarded. However, most images do not require the worker's action since the face detections are already fairly accurate. The 50 images include 3 gold standard images for quality control. These images have verified ground truth faces, but we intentionally show incorrect annotations for the workers to fix. The entire HIT resembles an action game. Starting with 2 lives, the worker will lose a life when making a mistake on gold standard images. In that case, they will see the ground truth faces (Fig. 7

---

[3]https://aws.amazon.com/rekognition

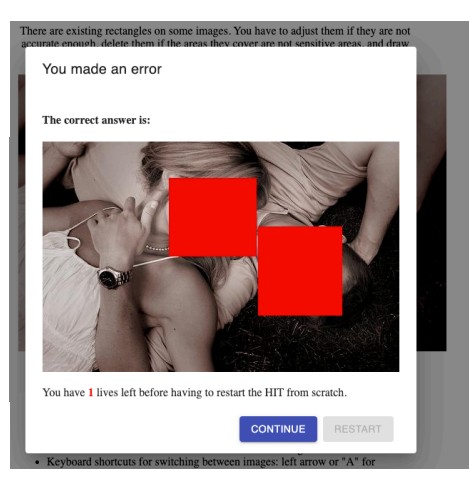

Figure 7: The UI for face annotation on Amazon Mechanical Turk. *Left*: The worker is given an image with inaccurate face detections. They correct the results by adjusting existing bounding boxes or creating new ones. Each HIT (Human Intelligence Task) have 50 images, including 3 gold standard images for which we know the ground truth answers. *Right*: The worker loses a life when making a mistake on gold standard images. They will have to start from scratch after losing both 2 lives.

*Right*) and the remaining lives. If they lose both 2 lives, the game is over, and they have to start from scratch at the first image. We found this strategy to improve annotation quality. We spent about $2500 on worker compensation. At this point, it is impossible to estimate the hourly wage accurately, since we have only the submit time of HITs but not the start time.

We did not distinguish NSFW areas from faces during crowdsourcing. Still, we conduct a study demonstrating that the final data contains only a tiny number of NSFW annotations compared to faces. The number of NSFW areas varies significantly across different ILSVRC categories. `Bikini` is likely to contain much more NSFW areas than the average. We examined all 1,300 training images and 50 validation images in `bikini`. We found only 25 images annotated with NSFW areas (1.85%). The average number for the entire ILSVRC is expected to be much smaller. For example, we found 0 NSFW images among the validation images from the categories in Table 1.

## B  FACE BLURRING METHOD

As illustrated in Fig. 8, we blur human faces using a variant of Gaussian blurring to avoid sharp boundaries between blurred and unblurred regions.

Let $\mathbb{D} = [0, 1]$ be the range of pixel values; $I \in \mathbb{D}^{h \times w \times 3}$ is an RGB image with height $h$ and width $w$ (Fig. 8 *Middle*). We have $m$ face bounding boxes annotated on $I$:

$$\{(x_0^{(i)}, y_0^{(i)}, x_1^{(i)}, y_1^{(i)})\}_{i=1}^m.^4 \tag{1}$$

First, we enlarge each bounding box to be

$$\left(x_0^{(i)} - \frac{d_i}{10}, y_0^{(i)} - \frac{d_i}{10}, x_1^{(i)} + \frac{d_i}{10}, y_1^{(i)} + \frac{d_i}{10}\right), \tag{2}$$

where $d_i$ is the length of the diagonal. Out-of-range coordinates are truncated to 0, $h - 1$, or $w - 1$.

Next, we represent the union of the enlarged bounding boxes as a mask $M \in \mathbb{D}^{h \times w \times 1}$ with value 1 inside bounding boxes and value 0 outside them (Fig. 8 *Bottom*). We apply Gaussian blurring to

---

[4]We follow the convention for coordinate system in PIL.

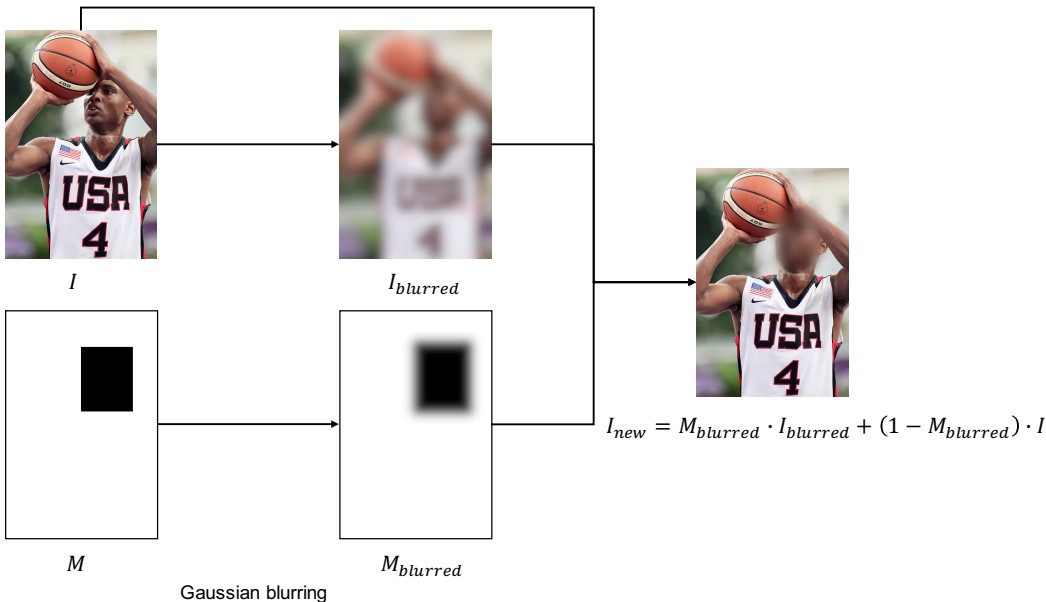

Figure 8: The method for face blurring. It avoids sharp boundaries between blurred and unblurred regions. $I$: the original image; $M$: the mask of enlarged face bounding boxes; $I_{new}$: the final face-blurred image.

both $M$ and $I$:

$$M_{blurred} = Gaussian\left(M, \frac{d_{max}}{10}\right) \tag{3}$$

$$I_{blurred} = Gaussian\left(I, \frac{d_{max}}{10}\right), \tag{4}$$

where $d_{max}$ is the maximum diagonal length across all bounding boxes on image $I$. Here $\frac{d_{max}}{10}$ serves as the radius parameter of Gaussian blurring; it depends on $d_{max}$ so that the largest bounding box can be sufficiently blurred.

Finally, we use $M_{blurred}$ as the mask to composite $I$ and $I_{blurred}$:

$$I_{new} = M_{blurred} \cdot I_{blurred} + (1 - M_{blurred}) \cdot I. \tag{5}$$

$I_{new}$ is the final face-blurred image. Due to the use of $M_{blurred}$ instead of $M$, we avoid sharp boundaries in $I_{new}$.

## C ORIGINAL IMAGES FOR TRAINING AND OBFUSCATED IMAGES FOR EVALUATION

We use obfuscated images to evaluate PyTorch models (Paszke et al., 2019)[5] trained on original images. We experiment with 5 different methods for face obfuscation: (1) blurring; (2) overlaying with the average color in the ILSVRC training data: a gray shade with RGB value $(0.485, 0.456, 0.406)$; (3–5) overlaying with red/green/blue patches.

Results in top-5 accuracy are in Table 7. Not surprisingly, face obfuscation lowers the accuracy, which is due to not only the loss of information but also the mismatch between data distributions in training and evaluation. Nevertheless, all obfuscation methods lead to only a small accuracy drop (0.7%–1.5% on average), and blurring leads to the smallest drop. The reason could be that blurring does not conceal all information in a bounding box compared to overlaying.

---

[5] https://pytorch.org/docs/stable/torchvision/models.html

Table 7: Top-5 accuracies of models trained on original images but evaluated on images obfuscated using different methods. *Original*: original images for validation; *Mean*: validation images overlaid with the average color in the ILSVRC training data; *Red/Green/Blue*: images overlaid with different colors; *Blurred*: face-blurred images; $\Delta_b$: Original minus blurred.

| Model | Original | Red | Green | Blue | Mean | Blurred | $\Delta_b$ |
|---|---|---|---|---|---|---|---|
| AlexNet (Krizhevsky et al., 2017) | **79.1** | 76.7 | 77.1 | 76.7 | 77.8 | 78.2 | 0.8 |
| GoogLeNet (Szegedy et al., 2015) | **89.5** | 87.9 | 88.2 | 87.9 | 88.3 | 88.7 | 0.9 |
| Inception v3 (Szegedy et al., 2016) | **88.7** | 86.7 | 87.0 | 86.6 | 87.2 | 87.7 | 0.9 |
| SqueezeNet (Iandola et al., 2016) | **80.6** | 78.6 | 79.0 | 78.5 | 79.4 | 79.7 | 0.9 |
| ShuffleNet (Zhang et al., 2018) | **88.3** | 86.6 | 86.8 | 86.6 | 87.0 | 87.4 | 1.0 |
| VGG11 (Simonyan & Zisserman, 2015) | **88.6** | 87.1 | 87.4 | 87.0 | 87.6 | 87.8 | 0.8 |
| VGG13 | **89.3** | 87.9 | 88.1 | 87.9 | 88.2 | 88.5 | 0.8 |
| VGG16 | **90.4** | 89.1 | 89.1 | 88.9 | 89.3 | 89.7 | 0.7 |
| VGG19 | **90.9** | 89.4 | 89.5 | 89.2 | 89.7 | 90.1 | 0.8 |
| MobileNet (Howard et al., 2017) | **90.3** | 88.9 | 89.1 | 88.9 | 89.2 | 89.5 | 0.8 |
| MNASNet (Tan et al., 2019) | **91.5** | 90.0 | 90.2 | 90.2 | 90.4 | 90.8 | 0.7 |
| DenseNet121 (Huang et al., 2017) | **92.0** | 90.7 | 90.8 | 90.7 | 91.0 | 91.3 | 0.7 |
| DenseNet161 | **93.6** | 92.5 | 92.5 | 92.3 | 92.8 | 93.0 | 0.6 |
| DenseNet169 | **92.8** | 91.6 | 91.7 | 91.6 | 91.9 | 92.2 | 0.6 |
| DenseNet201 | **93.4** | 92.2 | 92.3 | 92.0 | 92.3 | 92.7 | 0.7 |
| ResNet18 (He et al., 2016) | **89.1** | 87.5 | 87.6 | 87.5 | 87.8 | 88.3 | 0.8 |
| ResNet34 | **91.4** | 89.8 | 90.0 | 89.8 | 90.2 | 90.7 | 0.8 |
| ResNet50 | **92.9** | 91.7 | 91.8 | 91.5 | 91.8 | 92.2 | 0.7 |
| ResNet101 | **93.6** | 92.3 | 92.4 | 92.3 | 92.5 | 92.9 | 0.7 |
| ResNet152 | **94.1** | 92.9 | 93.0 | 92.9 | 93.1 | 93.4 | 0.6 |
| ResNeXt50 (Xie et al., 2017) | **93.7** | 92.5 | 92.6 | 92.4 | 92.8 | 93.0 | 0.7 |
| ResNeXt101 | **94.5** | 93.5 | 93.5 | 93.3 | 93.5 | 93.9 | 0.6 |
| Wide ResNet50 (Zagoruyko & Komodakis, 2016) | **94.1** | 92.9 | 93.0 | 92.9 | 93.1 | 93.4 | 0.7 |
| Wide ResNet101 | **94.3** | 93.2 | 93.3 | 93.1 | 93.4 | 93.7 | 0.6 |
| Average | **90.7** | 89.3 | 89.4 | 89.2 | 89.6 | 89.9 | 0.7 |

## D  OBFUSCATED IMAGES FOR TRAINING AND ORIGINAL IMAGES FOR EVALUATION

Vice versa, we also experiment with training on blurred images while evaluating on original images. This setting is practically relevant because models used in real-world products may be trained on privacy-preserved data but deployed in the wild without any obfuscation. Results are shown in Table 8. Similarly, training on blurred images lowers the accuracy by only a small amount (0.25%–1.04% in top-5 accuracy, with an average of 0.67%).

## E  DETAILS OF TRANSFER LEARNING EXPERIMENTS

**Image classification on CIFAR-10, SUN, and CelebA.** Object recognition on CIFAR-10 (Krizhevsky et al., 2009), scene recognition on SUN (Xiao et al., 2010), and face attribute classification on CelebA (Liu et al., 2015b) are all image classification tasks. For any model, we simply replace the output layer and finetune for 90 epochs. Hyperparameters are almost identical to those in Sec. 4, except that the learning rate is tuned individually for each model on validation data. Note that face attribute classification on CelebA is a multi-label classification task, so we apply binary cross-entropy loss to each label independently.

**Object detection on PASCAL VOC.** We adopt a FasterRCNN (Ren et al., 2015) object detector with a ResNet50 backbone pretrained on original or face-obfuscated ILSVRC. The detector is finetuned for 10 epochs on the trainval set of PASCAL VOC 2007 and 2012 (Everingham et al., 2010). It is then evaluated on the test set of 2007.

The system is implemented in MMDetection (Chen et al., 2019a). We finetune using SGD with a momentum of 0.9, a weight decay of $10^{-4}$, a batch size of 2, and a learning rate of $1.25 \times 10^{-3}$. The learning rate decreases by a factor of 10 in the last epoch.

Table 8: Validation accuracies on original ILSVRC images of models trained on original/blurred images. Training on blurred images lead to a small but consistent accuracy drop.

| Model | Top-1 accuracy (%) | | | Top-5 accuracy (%) | | |
|---|---|---|---|---|---|---|
| | Original training | Blurred training | Δ | Original training | Blurred training | Δ |
| AlexNet | **56.0** ± 0.3 | 55.3 ± 0.0 | 0.7 | **78.8** ± 0.1 | 78.0 ± 0.1 | 0.9 |
| SqueezeNet | **56.0** ± 0.2 | 54.9 ± 0.1 | 1.1 | **78.6** ± 0.2 | 77.6 ± 0.1 | 1.0 |
| ShuffleNet | **64.7** ± 0.2 | 63.7 ± 0.0 | 1.0 | **85.9** ± 0.0 | 85.1 ± 0.0 | 0.9 |
| VGG11 | **68.9** ± 0.0 | 67.9 ± 0.2 | 1.0 | **88.7** ± 0.0 | 87.9 ± 0.1 | 0.8 |
| VGG13 | **69.9** ± 0.1 | 69.0 ± 0.2 | 1.0 | **89.3** ± 0.1 | 88.6 ± 0.1 | 0.7 |
| VGG16 | **71.7** ± 0.1 | 70.6 ± 0.1 | 1.1 | **90.5** ± 0.1 | 89.8 ± 0.1 | 0.7 |
| VGG19 | **72.4** ± 0.0 | 71.2 ± 0.1 | 1.2 | **90.9** ± 0.1 | 90.1 ± 0.0 | 0.8 |
| MobileNet | **65.4** ± 0.2 | 64.0 ± 0.2 | 1.4 | **86.7** ± 0.1 | 85.6 ± 0.1 | 1.0 |
| DenseNet121 | **75.0** ± 0.1 | 74.1 ± 0.0 | 0.9 | **92.4** ± 0.0 | 91.8 ± 0.0 | 0.6 |
| DenseNet201 | **77.0** ± 0.0 | 76.5 ± 0.1 | 0.5 | **93.5** ± 0.0 | 93.2 ± 0.0 | 0.3 |
| ResNet18 | **69.8** ± 0.2 | 68.6 ± 0.2 | 1.1 | **89.2** ± 0.0 | 88.5 ± 0.1 | 0.7 |
| ResNet34 | **73.1** ± 0.1 | 72.0 ± 0.4 | 1.1 | **91.3** ± 0.0 | 90.6 ± 0.2 | 0.7 |
| ResNet50 | **75.5** ± 0.2 | 74.9 ± 0.1 | 0.6 | **92.5** ± 0.0 | 92.2 ± 0.1 | 0.3 |
| ResNet101 | **77.3** ± 0.1 | 76.6 ± 0.0 | 0.7 | **93.6** ± 0.1 | 93.2 ± 0.0 | 0.4 |
| ResNet152 | **77.9** ± 0.1 | 77.2 ± 0.2 | 0.7 | **93.9** ± 0.0 | 93.6 ± 0.0 | 0.4 |
| Average | **70.0** | 69.1 | 0.9 | **89.1** | 88.4 | 0.7 |

