# OpenReview forum: "A Study of Face Obfuscation in ImageNet"
_ICLR.cc/2022/Conference — ICLR 2022 Submitted_

### Official Review · Reviewer_U3FW · 2021-10-31

**Correctness:** 4
**Technical Novelty And Significance:** 2
**Empirical Novelty And Significance:** 2
**Recommendation:** 5
**Confidence:** 5

**Main Review:**

The main weakness:
1. I was confused by the statement in section (Part III, second paragraph) as to whether the face annotation process was manually filtered only for faces that were successfully detected (generated prediction boxes5) by Amazon Rekognition. Although I can infer from Appendix A Stage1 that only the data with successful detector predictions should be put into the AMT platform. Perhaps it would be better to include a brief description of Amazon Rekognition in the text.
2. And if some faces are not detected by Amazon Rekognition, how do you tackle this problem? This is an important problem for the privacy issue in this paper because this paper focuses on the privacy issue.
3. The novelty of this paper is limited reference to the proposed method in the paper.

The main strengths:
1. The experimental part of the paper is depicted in great detail and completely. Rigorous validation of the obfuscation approach is done on two different methods on 15 different models. And in the appendix, the detailed method of blurring, and the problems that may happen in the process of labeling are under clearer explanation.
2. The ethics of machine learning has been widely debated, with the issue of face privacy being of particular concern. There are many similar discussions, for example, there are some papers proposing to remove images associated with people from the database. The feasibility of obfuscation processing of faces proposed in this paper is a good way to minimize privacy issues without reducing the number of databases at the same time.

**Summary Of The Paper:**

The main concern addressed in this paper is the privacy problem that may result from images in ImageNet databases containing unexpected faces. The authors propose a two-step face filtering method. First, the authors use a detector called Amazon Rekognition to detect the ImageNet database. Then, the authors further optimize the detector output through the crowdsourcing platform Amazon Mechanical Turk (AMT) to reduce false positives and false negatives in automated detection. For the detected faces, the authors took two approaches, distinguishing between blurring and overlaying, and tested their effectiveness on different models separately. The accuracy of the two approaches was reduced by 0.9% on average compared to the original database on the ILSVRC classification challenge. And using the database that has blurred or covered the faces still maintains the transferability of the original database in the tests of downstream tasks.
Contribution：
1. The authors perform a very time-consuming and labor-intensive task for accurate labeling and filtering of faces in the ImageNet database and statistical analysis of the classes of faces contained in ImageNet.
2. The authors demonstrate experiments related to classification tasks and pre-trained model training using a database containing blurred or covered faces, proving that the theory is feasible and that the dropped accuracy is acceptable.
3. In terms of ethics, using blurred or covered face data for training can reduce privacy concerns. The study of the ImageNet database in terms of privacy can provide an important reference for subsequent databases

**Summary Of The Review:**

This paper has some contributions in exploring the ethicality of datasets, especially in the current very popular ImageNet database, but it exists some flaws (see weakness). The solutions and results in this paper are open sources and feasible, and this work will inspire subsequent exploration of privacy protection in publicly available datasets.

---

> ### Author Response · Authors · 2021-11-22
> **Individual Response to Reviewer U3FW**
>
> Thank you for your valuable feedback! Below we address your questions and concerns. Please feel free to post additional comments if you have further questions.
>
> ## Technical novelty
>
> Please see the common response above
>
>
> ## Do you only filter the faces identified by the face detector? What if some faces are not detected?
>
> This is definitely a concern we considered. In the human annotation step, workers are able to correct any errors in the automated face detector. Concretely, the workers (1) remove any false positive detections, (2) adjust the detected boxes to correct for localization errors and (3) add new detections as necessary to correct any false negatives. Please refer to Appendix A for details.

---

> > ### Comment · Reviewer_U3FW · 2021-11-24
> > **Final vote**
> >
> > First, thanks for your efforts on this work. It is still not well answered my question when some faces are not detected.  It still cannot clearly explain how to process the problems of the false positive/negative detections.  How do you use different detections to correct false negatives? I will still keep my rating.

---

> > > ### Author Response · Authors · 2021-11-24
> > > **Further Clarification**
> > >
> > > Thanks for the question. To clarify:
> > > * For false positives: the crowd worker can remove face bounding boxes detected by the face detector.
> > > * For false negatives: the crowd worker can draw new face bounding boxes.
> > >
> > > Hopefully that addresses your question. Please feel free to ask if anything remains unclear.

---

> > > > ### Comment · Reviewer_U3FW · 2021-11-25
> > > > **False Negatives**
> > > >
> > > > There are millions of images in ImageNet, it's impossible to check every image to find the face which is not detected by face detectors. And how to evaluate/judge the quality of the labels from crowd workers.
> > > >
> > > > Moreover, Imagenet is an impact work in computer vision. This paper is only done some little changes on it, the contribution is incremental. It cannot meet the high standard of this conference.

---

> > > > > ### Author Response · Authors · 2021-11-28
> > > > > **Clarification on the Face Annotation Process and Our Contributions**
> > > > >
> > > > > Thanks for engaging with our work and providing valuable feedback! We further address the questions about the face annotation process and the contributions of the work
> > > > >
> > > > > # Face Annotation Process
> > > > >
> > > > > Appendix A includes the full details of our face annotation method. Here we selectively cover a few points related to the reviewer’s question.
> > > > >
> > > > > ### Overview of the process
> > > > >
> > > > > 1. Given the 1.4 million images in ILSVRC, we use a face detector to detect faces automatically.
> > > > > 2. We ask crowd workers to correct the face detection results. They can remove false positives and draw new bounding boxes to cover false negatives.
> > > > >
> > > > >
> > > > > ### Feasibility
> > > > >
> > > > > Even though ILSVRC has more than 1.4 million images, our crowdsourcing pipeline is scalable enough to annotate faces on all images within a reasonable budget. As mentioned on page 17, we spent about $2500 on worker compensation.
> > > > >
> > > > > State-of-the-art face detectors are already quite accurate (Table 1 and Fig. 6). When checking the face detection results, the worker can quickly skip most images without editing them. Therefore, we were able to pack 50 images in each HIT (Human Intelligence Task). And annotating 1.4 million images needs less than 30K HITs.
> > > > >
> > > > >
> > > > > ### Quality control in crowdsourcing
> > > > >
> > > > > The paper discussed our quality control measures at the end of page 16. In summary, the 50 images in a HIT contain gold standard images for which we know the answer in advance. In order to finish the HIT, the worker has to achieve reasonable accuracy on gold standard images.
> > > > >
> > > > >
> > > > > ### Evaluating the quality of face annotations
> > > > >
> > > > > The paper has a dedicated subsection (page 4) on evaluating the quality of the annotations. In summary, we selected 20 categories on which the face detector makes more errors. Then we manually verified the face annotations on the validation images from these categories. Results (Table 1) show that our face annotations are of high quality. Among the 20 categories, we have on average 1.25 false positives and 0.95 false negatives per 50 images. Further, the overall accuracy on the entire ILSVRC is much higher as these categories are selected deliberately to be error-prone.
> > > > >
> > > > >
> > > > > # Our Contributions
> > > > >
> > > > > We agree that ImageNet is a high-impact work (with > 34,000 citations to date) and that our contributions may not be on the scale of ImageNet. Nevertheless, we respectfully disagree that the contributions are incremental. Face annotations are important for studying privacy in visual data (e.g., Oh et al. 2016 and Ren et al. 2018). ImageNet is an important computer vision dataset, but we are not aware of existing large-scale face annotations on it. We fill this gap and provide useful data for the computer vision community to study the privacy aspects of their models. Further, we release ImageNte pre-trained models on our face-obfuscated data, which can be directly plugged into existing pipelines and systems that previously relied on ImageNet pre-training.
> > > > >
> > > > > In addition to the dataset contribution, we are the first to investigate the effects of privacy-aware face obfuscation on large-scale visual recognition. Our extensive experiments demonstrate that training on face-obfuscated images does not significantly compromise accuracy. Therefore, we advocate for face obfuscation to become a standard step in future dataset creation efforts. We will release our annotation interfaces and crowdsourcing scripts, enabling this effort.
> > > > >
> > > > >
> > > > > * Seong Joon Oh, Rodrigo Benenson, Mario Fritz, and Bernt Schiele. Faceless person recognition: Privacy implications in social media. In European Conference on Computer Vision, 2016.
> > > > > * Zhongzheng Ren, Yong Jae Lee, and Michael S Ryoo. Learning to anonymize faces for privacy preserving action detection. In European Conference on Computer Vision, 2018.

---

### Official Review · Reviewer_6ZXN · 2021-11-03

**Correctness:** 4
**Technical Novelty And Significance:** 2
**Empirical Novelty And Significance:** 3
**Recommendation:** 6
**Confidence:** 3

**Main Review:**

The main strength of the paper is to address the privacy issues of ImageNet and is to provide an alternative face obfuscated version.
In addition, the authors conduct very thorough experiments across different tasks and different architecture for the study of the performance influence with the obfuscated dataset. It shows new dataset still is effective for transfer learning for various vision task with few performance drop. However, the weakness is that the main part of the paper is to examine the performance influence of different settings and is of limited technical novelty. For the verification of some downstream tasks, the coverage of tasks is not enough.

**Summary Of The Paper:**

The paper mainly discuss the privacy issue for human for the widely used ImageNet dataset and how to handle them.
In addition, the paper does a very detailed empirical experiments to study the performance influence for various tasks, including object recognition, scene recognition, face attribute, and object detection if all the faces in the ImageNet are obfuscated.

**Summary Of The Review:**

Although the paper mainly focuses on providing plenty of empirical results to evaluate the influence of using the face obfuscated ImageNet dataset and is of limited novelty, it provides a lot of insights to show the effectiveness and feasibility of privacy preserving ImageNet.

I have few concerns of the selection of transferring tasks. For example, the resolution of CIFAR-10 is only 32x32 and only for 10 classes. Similarly, Pascal VOC is also relatively small and easy dataset as compared with COCO or other recently released object detection dataset.
Most of the images in the CelebA dataset are in frontal pose and have much fewer variations than other unconstrained face dataset, like IJB-C, etc. Since these datasets are relatively simpler than others which are more close to real-world scenarios, I wonder if the same experimental results and findings are still valid for harder datasets with more variations.

---

> ### Author Response · Authors · 2021-11-22
> **Individual Response to Reviewer 6ZXN**
>
> Thank you for your valuable feedback! Below we address your questions and concerns. Please feel free to post additional comments if you have further questions.
>
> ## Technical novelty
>
> Please see the common response above
>
>
> ## The datasets in the transfer learning experiments are too simple.
>
> We choose simpler datasets such as CIFAR-10, PASCAL VOC, and CelebA deliberately, because they are more likely to benefit from ImageNet pretraining. When studying the effect of face obfuscation on ImageNet’s utility as a pretraining dataset, we’d like to use datasets on which ImageNet pretraining makes a large difference. For example, we include PASCAL VOC but not COCO since prior work has shown that ImagetNet pretraining is not necessary for obtaining good performance on COCO [F].
>
>
> [F] He, Kaiming, Ross Girshick, and Piotr Dollár. "Rethinking ImageNet pre-training." CVPR, 2019.

---

> > ### Comment · Reviewer_6ZXN · 2021-11-24
> > **Thanks for the reply**
> >
> > Thanks for the reply of the authors. I truly appreciate the efforts done by the work for addressing the privacy issue of the ImageNet dataset. However, I would still want to see the influence and difference w/ and w/o face obfuscation for  larger and more complex datasets which can get benefit from ImageNet pretraining. I will still keep my rating.

---

### Official Review · Reviewer_7itG · 2021-11-04

**Correctness:** 4
**Technical Novelty And Significance:** 1
**Empirical Novelty And Significance:** 2
**Recommendation:** 5
**Confidence:** 4

**Main Review:**

Strengths:
* The main contribution of this work is that it provided empirical evidence on the effect of face obfuscation on the ImageNet dataset. Through comprehensive experiment, the authors showed that face obfuscation does not decrease the utility of the dataset.
* Another contribution that should not be overlooked is that the authors annotated all faces in ImageNet in a semi-automatic manner and promised that they will make the annotations publicly available to other researchers.
* The paper is very well written and includes a lot of details on the experiment protocol. Thus It should be straightforward for other researchers to reproduce the results and to extend the study.

Weaknesses:
* Since blur and cutout are commonly used data augmentation techniques, it is to be expected that face obfuscation would not have a big impact to visions tasks that have little to do with faces. Although it is commendable that this is now shown empirically though the study, this work also does not bring interesting new insights into the topic.
* The authors showed that categories that are closely related to faces are indeed affected more by face obfuscation. This paper would be more interesting (from a technical point of view) if the authors could additionally investigate into methods for alleviating such impact.

**Summary Of The Paper:**

This paper presents an empirical study on the effect of face obfuscation in the ImageNet dataset. The main conclusion is that face obfuscation does not decrease the utility of the dataset. Specifically, the authors showed that various networks trained on the obfuscated dataset only experienced small accuracy drop on the image classification task. The authors also discussed the impact on different categories, showing that face obfuscation hurt more to the object categories that are more closely related to faces (i.e., the bounding boxes of which overlap more with faces). Last but now least, experiments has been conducted to show that face obfuscation also does not have a significant impact on the transferability of the features learned from the new dataset. All these conclusions are inline with intuitions since ImageNet is not primarily focused on human activities / faces.

**Summary Of The Review:**

This paper is very well written and it provides empirical evidences to support the intuition that face obfuscation does not decrease the utility of the ImageNet dataset. However, my main concern is that the paper has no technical novelty and it also does not bring sufficiently new insights to the community.

---

> ### Author Response · Authors · 2021-11-22
> **Individual Response to Reviewer 7itG**
>
> Thank you for your valuable feedback! Below we address your questions and concerns. Please feel free to post additional comments if you have further questions.
>
> ## Technical novelty
>
> Please see the common response above
>
> ## Since blur and cutout are commonly used data augmentation techniques, it is expected that face obfuscation has a marginal impact on vision tasks that have little to do with faces.
>
> Privacy-aware face obfuscation is completely different from data augmentation such as cutout. Cutout augments the training images by applying random masks. Since masks are at random locations, the masked area may become visible when the same image appears again in another epoch. Therefore, cutout does not necessarily remove any information from the training data. In contrast, face obfuscation systematically removes information in the face bounding boxes, which may reasonably lead to the accuracy drop.
>
> Also, the results of our experiments cannot be trivially inferred. Even if a vision task has little to do with faces, it is possible that models learn to exploit spurious face-related visual cues for making predictions. For example, prior works have shown that models rely on gender cues to perform action recognition (Hendricks et al., 2018).

---

### Author Response · Authors · 2021-11-22
**Common Response**

We thank all reviewers for their thoughtful comments. They agree that our thorough empirical experiments (6ZXN,U3FW) provide a lot of insights (6ZXN) and that our paper is well-written (7itG). We are especially encouraged that reviewers think our work provides important reference for subsequent data collection practices (U3FW) and our face annotations is a valuable service to the community (7itG, U3FW)

Reviewers’ concerns focus on technical novelty. They point out that our paper only analyzes existing methods/datasets without proposing any novel techniques. This is a point that we totally agree with. However, we respectfully but wholeheartedly disagree that a paper must have algorithmic contributions to be important and impactful. Many great papers published in machine learning and computer vision do not focus on algorithmic contributions. Instead, they focus on analyzing existing methods/datasets ([A], [B], [C]), or provide a service to the community (e.g., by constructing datasets [D] [E]).

* [A] Torralba and Efros. "Unbiased look at dataset bias." CVPR, 2011.
* [B] Shankar et al. "Evaluating machine accuracy on imagenet." ICML, 2020.
* [C] Tsipras et al. "From imagenet to image classification: Contextualizing progress on benchmarks." ICML, 2020.
* [D] Deng et al. "Imagenet: A large-scale hierarchical image database." CVPR, 2009.
* [E] Lin et al. "Microsoft coco: Common objects in context." ECCV, 2014.

---

### Decision · Program_Chairs · 2022-01-20

**Decision:**

Reject

**Comment:**

This paper received 3 quality reviews, with 2 rated 5 and 1 rated 6. While the reviewers recognize the various contributions and insights made by this work, it was also pointed out that this work lacks technical novelty. The authors agreed with this concerns and argued that this work provides a service to the community, citing imageNet and COCO papers. The AC agrees with the contribution and major concerns. Furthermore, the AC would like to point out that in term of the level of efforts, this work might not be on par with the imageNet and COCO. All things considered, the AC believes that this work is not ready for publication at its current form, and hence recommend rejection.